

# Association between changes in social capital and mental well-being among older people in China

Huihui Wang[1,*], Jingni Zhang[2,*], Zhenfan Yu[1], Naifan Hu[1], Yurun Du[1], Xiaoxue He[1], Degong Pan[1], Lining Pu[1], Xue Zhang[1] and Jiangping Li[1,3]

[1] Department of Epidemiology and Health Statistics, School of Public Health and Management, Ningxia Medical University, Yinchuan, Ningxia Hui Autonomous Region, China
[2] Department of Science and Education, Qinghai Provincial People's Hospital, Xining, Qinghai Province, China
[3] Key Laboratory of Environmental Factors and Chronic Disease Control, Ningxia Medical College, Yinchuan, Ningxia Hui Autonomous Region, China
[*] These authors contributed equally to this work.

## ABSTRACT

**Background**. The mental well-being of older people has become social concern under aging times in China. Social capital has been linked to mental well-being. Our aims were to explore how social capital and the state of mental well-being of older people were changing and what the relationship between them was.

**Methods**. Data were from six waves of the China Family Panel Studies that were conducted between 2010 and 2020, and a total of 1,055 participants aged 60 and over were included in the analysis. The Generalized Estimated Equation model (GEE) was used to clarify the long-term relationship, and to use GEE we first defined how time points were related, in other words, an appropriate working correlation structure was supposed to choose. Therefore, correlation coefficient between measurements at two time points was calculated to choose the exchange structure. All the analyses were performed in the statistical software Stata 15.0.

**Results**. The mental well-being of older people has deteriorated over time, especially we found that between 2014 and 2016, the mental well-being of older people plummeted. In addition, cognitive social capital was positively correlated with mental well-being, while structural social capital was inverse.

**Conclusions**. Policymakers are supposed to take into account the long-term impact of cognitive and structural social capital on the mental well-being of older people and to provide them with projects aimed at increasing cognitive social capital and turning the pressure of structural social capital into a source of happiness in life.

Corresponding author
Jiangping Li, lijp@nxmu.edu.cn

## INTRODUCTION

Owing to remarkable increases in life expectancy and decreases in birth rates, the world population is ageing at an unprecedented rate (*Felez-Nobrega et al., 2021*). In recent decades, as a rapid change of social in China, the problem of ageing population has become more acute than other countries (*Zhong et al., 2017*). According to *Chang (2021)* by 2019,

254 million people were aged 60–64 and another 176 million were aged 65 plus, and by 2040, an estimated 402 million people (28% of the total population) will be over the age of 60. In addition, a research has shown that older people are more likely to suffer mobility disabilities, chronic pain, weakness, or decline in their socio-economic status, and all of these stressors could lead to isolation, loneliness or psychological distress (*Nyqvist & Nygård, 2013*). Further, statistics have shown that approximately 20% of adults aged 60 and over suffer from a mental disorder (*Grolli et al., 2021*). The problem of mental well-being in older people has brought great burden to the society, public health and medical system, and needs pay more attention to it.

In recent years, social capital has been become the hot topic of scientific research, and it is a new academic term coined in the social sciences in the early 20th century. The most widely cited definition of social capital within health research is the one by Robert D. Putnam who suggests that social capital is a shared property based on community activities and not of individuals alone (*Nyqvist et al., 2013*). Specifically, defined as "the norm of social networking and reciprocity", communities deemed rich in social capital are made up of individuals who exhibit a high degree of general trust, a high degree of sociability and civic engagement and high levels of universal reciprocity (*Kiechel, 2000*). In this research, we considered individual-level social capital as a multidimensional concept, which can be measured by cognitive and structural dimensions (*Agampodi et al., 2015*). Structural social capital, which mainly refers to the objective social structure such as social organization and network; while cognitive social capital, also called cultural social capital, refers to norms, values, attitudes, beliefs, trust, reciprocity and other psychological processes (*Bowling et al., 2002*; *Islam et al., 2006*).

Mental well-being has been shown to be associated with social capital (*Giordano & Lindstrom, 2011*; *Nyqvist et al., 2013*; *Chipps & Jarvis, 2016*; *Flores et al., 2017*; *Ehsan et al., 2019*). There is strong evidence that, on average, the impact of social capital on mental well-being is positive. Studies have shown a higher level of social capital is related to fewer depressive symptoms (*Howley, Neill & Atkinson, 2015*; *Simons et al., 2020*; *Cao et al., 2022*). This may be due to communities with high levels of social capital are more likely to discourage behaviors such as drinking, smoking, and crime, and even promote mentally healthier behaviors, such as regular exercise (*Giordano & Lindstrom, 2010*; *Tennison et al., 2010*). Interestingly, however, some research showed a negative relationship, and individual studies also found insignificant relationship (*Almedom, 2005*; *Ehsan et al., 2019*). This may be because of excessive informal control, and higher social capital can entail a restriction of freedom, resulting in greater psychological stress (*Portes, 1998*). In addition, multiple studies have shown that the relationship between cognitive and structural social capital and mental well-being varies across country and study designs (*Ehsan & De Silva, 2015*; *Coll-Planas et al., 2017*; *Ehsan et al., 2019*). However, previous research in older people were conducted more frequently in high-income countries, and were often based in the USA, the UK, or Scandinavian countries, mainly in Caucasian older people (*Coll-Planas et al., 2017*). Moreover, a majority of studies were descriptive and cross-sectional in design (*Agampodi et al., 2015*). There is an urgent need to longitudinal designs because of stronger

causal associations than cross-sectional, which can provide stronger evidence for the relationship between social capital and mental well-being.

Therefore, in this article, we designed a longitudinal study to explore how social capital and the state of mental well-being of older people were changing and what the relationship between them was from 2010 to 2020.

## METHODS

### Study participants

Data (2010∼2020) from the China Family Panel Studies (CFPS), with follow-up every two years, were used. The CFPS was launched by the China Research Center for Social Sciences at Peking University to track the changes in Chinese society, economy, population, education and health by collecting data at the individual, family and community levels. CFPS is a research projects involving people. In order to ensure that the rights and interests of the respondents are protected to the greatest extent, the ethics review is regularly submitted to the "Peking University Biomedical Ethics Committee", and the corresponding data collection work is carried out when the ethics review is approved.

In this study, we selected people aged 60 and over and designed a longitudinal study. After excluding missing and lost to follow-up individuals, we obtained the full panel data with a total of 1,055 participants.

### Measurement of mental well-being

Mental well-being was the dependent variable of the study, and the score of mental state was taken as the index to measure the mental well-being of older people. There were three mental well-being scales, CESD-20, CESD-8 and K6 in CFPS, which had shown good reliability and validity in the previous studies (*Turvey, Wallace & Herzog, 1999*; *Dai & Gu, 2021*). CESD-20, called the Center for Epidemiological Studies Depression scale, was developed in 1977 and used to measure depressive symptoms in the general population (*Kim & Lee, 2013*), and the CESD-8 is abridged version of the CESD-20. The Kessler Psychological Distress scale (K10) developed by Kessler at the University of Michigan was able to assess the risk of mental well-being in a population, and the K6 is a subset of the K10 (*Cornelius et al., 2013*). Six questions from these scales were selected, namely "how often do you feel emotionally depressed, nervous, restless and difficult to do anything, have no hope for the future, and think life has no meaning in the past month". Four answers, 0 = never, 1 = some times, 2 = often, and 3 = most of the time. The Cronbach's alpha for these six questions is 0.7748 and the kmo value is 0.8514, which is regarded as satisfactory and acceptable (*Taber, 2017*; *de Barros Ahrens, Lirani & De Francisco, 2020*). Respondents' mental well-being scores were calculated by adding up the scores for each question. The variable on mental well-being was used in the analysis as a categorical variable ($\geq 3$ (code 0) or <3(code 1)).

### Measurement of social capital

According to *Harpham, Grant & Thomas (2002)*, "Institutional linkages", "Family and friends connections" and "Proactivity in social context" were used to measure structural

social capital, and "Value of life", "Feeling of trust and safety" and "Tolerance of diversity" were used to describe cognitive social capital. Although the exact same data was not available in CFPS, we used the similar social capital variables.

The way to assess structural social capital was to ask respondents whether they have pension and medical insurance, whether they have a job, who usually take care of them when they are unwell, and where they usually go for medical treatment when they are ill. Furthermore, cognitive social capital was measured by asking respondents about their satisfaction with medical conditions, satisfaction of medical level and life satisfaction, evaluation of the local municipal government, and confidence in their future. The Cronbach's alpha for these questions is 0.6108 and the kmo value is 0.6192, which is regarded as satisfactory and acceptable (*Taber, 2017*; *de Barros Ahrens, Lirani & De Francisco, 2020*). The total scores were calculated by adding up the scores for each question, and the higher the score, the more structural and cognitive social capital they had.

### Control variables

The following individual-level covariates were considered and controlled in the analysis: residence, age, sex, highest level of education achieved and marital status.

### Working correlation structure

Each independent variable was run against the dependent variable using Generalized Estimating Equations (GEE). Reasoning behind this choice of model was twofold: firstly, repeated observations within the same subject are not independent of each other. Secondly, the dependent variable is discrete.

To use GEE we must first define how time points are related, in other words, an appropriate working correlation structure were supposed to choose. The study has shown that no matter which structure is chosen, the result of GEE analysis is stable (*Liang & Zeger, 1986*). However, another research has deemed that the conclusion that there is a little connection between the results of GEE analysis and the wrong choice of correlation structure is only applicable to the binary classification variables (*Zeger, 2010*). Hence, we calculated the correlation coefficient between measurements at two time points to determine which working correlation structure to choose.

### Statistical analysis

Based on the characteristics of the data, the GEE of two-classification was chosen. The model is as follows:

$$\text{logit}\left(\text{E}\left(Y_{ij}\right)\right) = \beta_0 + \beta_1 X_{ij} + \beta_2 X_{ij} + \ldots + \beta_n X_{ij}$$

$i = 1, \ldots, 1055$, $j = \text{T}(2010), \ldots, \text{T}(2020)$, $\text{logit}(\cdot)$ is called the join function; $Y_{ij}$ is the mental well-being of older people for subject i at time j; $X_{ij}$ is the explanatory variable of $Y_{ij}$.

All the analyses were performed in the statistical software Stata 15.0, $P < 0.05$, and the differences were statistically significant.

## RESULTS

### Baseline characteristics

A total of 1,055 participants aged 60 and over were included in the analysis. A total of 55.1% were from rural areas, and 88.6% were aged 60 to 70. More than half of the subjects were male and 62.6% had an education level of primary school or less, and 89.9% were married (Table 1).

### Working correlation structure

Table 2 shows the outcome of the working correlation structure of outcome variable, we decided to select the exchange structure after thinking over the outcome of analysis. This is where the correlation between observations at two time points is equal for any two time points.

### Trends in the mental well-being of older people

Table 3 shows that the mental well-being of older people has deteriorated over time ($P < 0.001$, OR = 0.844). Figure 1 shows trends in the number of people with better mental well-being from 2010 to 2020, and we found that between 2014 and 2016, the mental well-being of older people plummeted.

### Data analysis

Regarding the confounding factors (Table 3), older people living in towns had better mental well-being than those living in the countryside ($P < 0.001$, OR = 1.695). And women were 0.555 times more likely to have better mental well-being than men. Furthermore, older age was associated with poorer mental well-being ($P < 0.001$, OR = 0.941); Education levels and marital status were positively correlated with good mental well-being ($P < 0.001$, OR =1.807; $P < 0.001$, OR = 1.654).

Multi-factor GEE analyses were used after controlling confounding factors. The cognitive social capital was positively correlated with good mental well-being ($P < 0.001$, OR = 1.050), however, there was inverse relationship between structural social capital and mental well-being ($P < 0.001$, OR = 0.939).

## DISCUSSION

The purpose of this study was to explore the relationship between changes of structural and cognitive social capital and the mental well-being of older people over time. The results revealed that structural and cognitive social capital were correlated with mental well-being.

In the present study, we found a dramatic drop in mental well-being scores among older adults between 2014 and 2016, and believed this is due to an increase in the number of households. On January 1, 2016, Population and Family Planning Law of the China was officially implemented the two-child policy fully opened. According to the National Bureau of Statistics of the People's Republic of China, in 2016, Chinese birth rate was 12.95%, up 7% from 2015. Furthermore, the increase of grandchildren will lead to the inadequacy of family resource, which may reduce the cognitive social capital (*Hansen, 2011*), thus perhaps affecting the mental well-being of family members.

**Table 1 Frequencies of variables expressed as percentages (%) of stratified by psychological status at baseline.**

| Variables | Poor mental well-being | Good mental well-being | Total |
|---|---|---|---|
| Residence | | | |
| Rural | 255(65.2%) | 326(49.1%) | 581(55.1%) |
| Urban | 136(34.8%) | 338(50.9%) | 474(44.9%) |
| Age | | | |
| 60~70 | 348(89.0%) | 587(88.4%) | 935(88.6%) |
| 70~80 | 43(11.0%) | 75(11.3%) | 118(11.2%) |
| 80~90 | 0(0.0%) | 2(0.3%) | 2(0.2%) |
| Gender | | | |
| Male | 180(46.0%) | 398(59.9%) | 578(54.8%) |
| Female | 211(54.0%) | 266(40.1%) | 477(45.2%) |
| Education levels | | | |
| Primary school or less | 288(73.7%) | 372 (56.0%) | 660(62.6%) |
| Junior high | 94(24.0%) | 257(38.7%) | 351(33.3%) |
| Senior high | 9(2.3%) | 23(3.5%) | 32(3.0%) |
| Undergraduate or higher | 0(0.0%) | 12(1.8%) | 12(1.1%) |
| Marital status | | | |
| Not married | 6(1.5%) | 3(0.5%) | 9(0.9%) |
| Divorced | 5(1.3%) | 2(0.3%) | 7(0.7%) |
| Widowed | 48(12.3%) | 43(6.5%) | 91(8.6%) |
| Married | 332(84.9%) | 616(92.8%) | 948(89.9%) |

**Table 2 The working correlation structure of the outcome variable.**

| | Y(2010) | Y(2012) | Y(2014) | Y(2016) | Y(2018) | Y(2020) |
|---|---|---|---|---|---|---|
| Y(2010) | 1.0000 | | | | | |
| Y(2012) | 0.2498 | 1.0000 | | | | |
| Y(2014) | 0.3269 | 0.3277 | 1.0000 | | | |
| Y(2016) | 0.2225 | 0.2010 | 0.2327 | 1.0000 | | |
| Y(2018) | 0.2410 | 0.3238 | 0.3460 | 0.2407 | 1.0000 | |
| Y(2020) | 0.2213 | 0.2569 | 0.2831 | 0.1936 | 0.3512 | 1.0000 |

**Notes.**
Y refers to the mental well-being of older people (dependent variable).

The demographic factors showed significant correlation with the mental well-being of older people. In line with the previous study (*Weissman et al., 1996*), this study also showed a gender difference, with women having worse mental well-being than men. What this likely is due to women is at greater risk of gender-based violence and therefore bear more psychological pressure (*Kiely, Brady & Byles, 2019*). Furthermore, consistent with the Nyqvist study (*Nyqvist & Nygård, 2013*), our results suggested that the mental well-being of older people tended to worsen with age. Interestingly, however, some studies suggest that older and younger adults have better mental well-being than middle-aged adults (*Nyqvist et al., 2013*). One British study even suggests that aging is a protective factor for mental

**Table 3   The effect of all variables on the mental well-being of older people.**

| Covariates | Coef. | Robust Std. Err | Z | P | OR | 95% CI |
|---|---|---|---|---|---|---|
| Time | −0.170 | 0.013 | −12.560 | <0.001 | 0.844 | 0.822~0.867 |
| Residence | 0.528 | 0.074 | 7.090 | <0.001 | 1.695 | 1.465~1.962 |
| Gender | −0.589 | 0.076 | −7.790 | <0.001 | 0.555 | 0.479~0.644 |
| Age | −0.061 | 0.006 | −10.470 | <0.001 | 0.941 | 0.930~0.952 |
| Education levels | 0.592 | 0.067 | 8.780 | <0.001 | 1.807 | 1.583~2.062 |
| Marital status | 0.503 | 0.083 | 6.030 | <0.001 | 1.654 | 1.404~1.974 |
| Structural social capital[*] | −0.063 | 0.015 | −4.110 | <0.001 | 0.939 | 0.912~0.968 |
| Cognitive social capital[*] | 0.048 | 0.008 | 5.760 | <0.001 | 1.050 | 1.032~1.067 |

**Notes.**
   [*]Confounding factors were controlled such as age, sex, and residence, education levels, marital status.

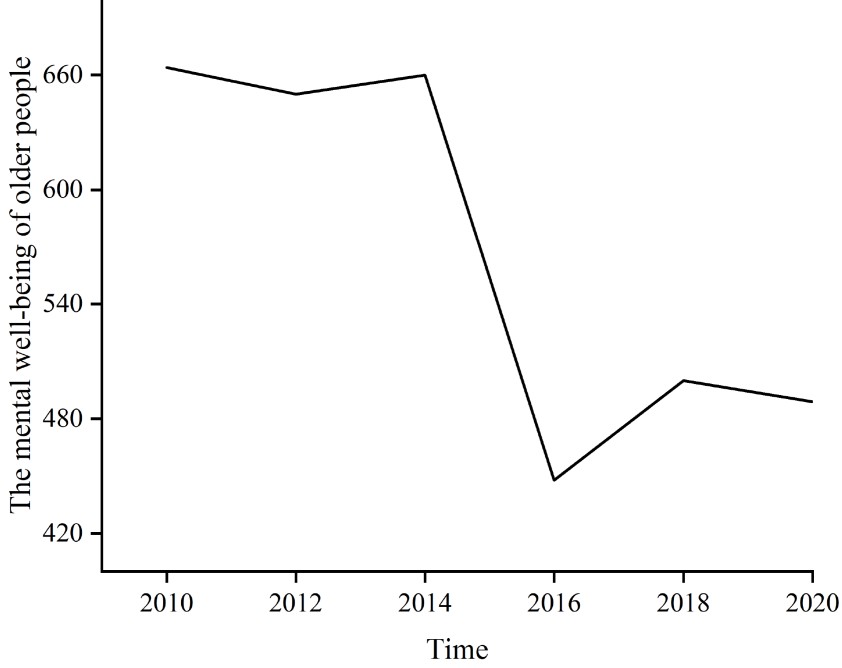

**Figure 1   Trends in the number of people with better mental well-being from 2010 to 2020.**

well-being (*Giordano & Lindstrom, 2011*). We believe that the inconsistent results may be due to subjects in the British study transitioning from middle to old age. We also found higher education levels older adults got, the better their mental well-being, which was consistent with previous cross-sectional studies (*Ajrouch, 2007*). And the educational level can also measure the lifetime economic status of older people, and the lower economic status would increase the risk of isolation, bringing great pressure to the mental well-being in older people (*Van Groenou & Van Tilburg, 2003*). In addition, we found that older people in towns had better mental well-being than countryside. This may be due to the low level of education in rural China (*Zhang, Xu & Lu, 2019*), and they may not have the knowledge to deal with mental well-being issues. This shows that education is an important

indicator affecting the mental well-being. In addition, we also found that marriage had a positive effect on the long-term mental well-being. Some studies suggest that married seniors have better mental states than unmarried seniors (*Chen, Waite & Lauderdale, 2015*) because marriage may provide some benefits (such as spousal care, support, and companionship) (*Hagedoorn et al., 2006*).

Our research also found an inverse relationship between structural social capital and mental well-being. However, previous studies had yielded mixed results (*De Silva et al., 2005*; *Cao et al., 2015*). A study in China revealed that there was a positive correlation (*Liang et al., 2020*). They assumed that structural social capital could induce more collective actions, which hold promise for improving the health and well-being of the Chinese population by promoting healthy behavior. However, a longitudinal study in Korea suggested the structural social capital of poor older women was low on the protective aspects of health outcomes (*Park, 2017*). In addition, some studies suggested that structural social capital was perhaps protective against mental well-being in some countries and not others in older people (*Fujiwara & Kawachi, 2008*; *Wang et al., 2022*). We assume this may be caused by cultural differences between different countries (*Agampodi et al., 2015*).

In recent years, with the rapid development of science and technology, life has been become more and more convenient, but the response and acceptance ability of older people has declined (*Van Groenou & Van Tilburg, 2003*). Therefore, they are at a disadvantage in the application of new resource such as the Internet. This undoubtedly puts a certain amount of pressure on their mental well-being. Also, in Chinese culture, more and more older people don't want to cause trouble to their children. For example, most older people can't use self-service machines to withdraw money, so endowment insurance may be received by their children, and then become a resource for the children rather than older people themselves. In addition, when older people are sick, their children may drop work at hand to take care of them in the hospital. This places a great psychological burden because they see themselves as a burden on their children. The division of departments in large hospitals is becoming more and more detailed, and it is difficult for older people to find the corresponding position smoothly, which can make them feel inferior and useless. Instead, they were able to cope with the small clinics in the villages and the township health centers.

Furthermore, previous studies had shown a positive correlation between cognitive social capital and mental well-being in older adults (*Bowling et al., 2002*; *Theurer & Wister, 2009*; *Dai & Gu, 2021*; *Wang et al., 2022*) and our longitudinal study showed the same results. Older people's satisfaction with life, satisfaction with health care, confidence in the future and evaluation of the local municipal government indicate how much stress they feel in their lives, from stress to mental well-being, probably through the hypothalamus-pituitary-adrenal (HPA) axis (*Itoi & Sugimoto, 2010*; *Tennison et al., 2010*; *Giordano & Lindstrom, 2011*). HPA axis dysfunction, a response to perceive stressors, plays an important role in mood. On the contrary, if there is no pressure or less pressure, that is, higher cognitive social capital, it perhaps prompts the mental well-being of older people.

Based on the above discussion, policy makers should formulate a set of policy systems applicable to older people while improving the local technological level and living standards, so as not to let technology become a stumbling block to the happy life of older people.

## LIMITATIONS

Since there is no gold standard for measuring social capital, we chose a comprehensive questionnaire to measure them. Furthermore, the six-year follow-up mental well-being scales in the CFPS were not all the same, and we picked six similar questions in each year. However, the interaction between structural social capital and cognitive social capital were not considered, which may lead to error.

## CONCLUSION

This study provided evidence for the long-term relationship between social capital and mental well-being of older people using GEE with the exchange structure. Specifically, cognitive social capital and mental well-being are positively correlated meaning that satisfaction and self-confidence of older people boost their mental well-being. However, there was a negative correlation between structural social capital and their mental well-being. Thus, we assume that some resources perhaps are regarded as a burden rather than benefits for older people. From the above, policymakers are supposed to take into account the long-term impact of cognitive and structural social capital on the mental well-being of older persons and to provide them with projects aimed at increasing cognitive social capital and turning the pressure of structural social capital into a source of happiness in life.

## ACKNOWLEDGEMENTS

We want to give a special acknowledgement to CFPS for providing real and reliable data for academic research.

### Funding

This work was supported by the Key R&D Program of Ningxia Autonomous Region (2022BEG03106). The funders had no role in study design, data collection and analysis, decision to publish, or preparation of the manuscript.

### Grant Disclosures

The following grant information was disclosed by the authors:
The Key R&D Program of Ningxia Autonomous Region: 2022BEG03106.

### Competing Interests

The authors declare there are no competing interests.

### Author Contributions

- Huihui Wang performed the experiments, analyzed the data, prepared figures and/or tables, authored or reviewed drafts of the article, and approved the final draft.
- Jingni Zhang conceived and designed the experiments, performed the experiments, analyzed the data, prepared figures and/or tables, authored or reviewed drafts of the article, and approved the final draft.

- Zhenfan Yu performed the experiments, authored or reviewed drafts of the article, and approved the final draft.
- Naifan Hu conceived and designed the experiments, authored or reviewed drafts of the article, and approved the final draft.
- Yurun Du conceived and designed the experiments, authored or reviewed drafts of the article, and approved the final draft.
- Xiaoxue He conceived and designed the experiments, authored or reviewed drafts of the article, and approved the final draft.
- Degong Pan analyzed the data, prepared figures and/or tables, and approved the final draft.
- Lining Pu performed the experiments, prepared figures and/or tables, and approved the final draft.
- Xue Zhang analyzed the data, prepared figures and/or tables, and approved the final draft.
- Jiangping Li conceived and designed the experiments, performed the experiments, authored or reviewed drafts of the article, and approved the final draft.

## Ethics

The following information was supplied relating to ethical approvals (i.e., approving body and any reference numbers):

This study was approved by The Peking University Biomedical Ethics Committee(IRB0000105214010).

## Data Availability

The raw measurements and code are available in the Supplemental Files.

## Supplemental Information

Supplemental information for this article can be found online at http://dx.doi.org/10.7717/peerj.13938#supplemental-information.

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
