# Peer review of "Association between changes in social capital and mental well-being among older people in China"

_PeerJ, doi:10.7717/peerj.13938_

## Round 0.1 · original submission · Minor Revisions

I have now received the reviewers' comments on your manuscript. They have suggested some minor revisions to your manuscript. Therefore, I invite you to respond to the reviewers' comments and revise your manuscript.

Reviewer 1 ·

Basic reporting

Abstract:
The abstract of the article is very short (less than 250 words). In the method section, you should mention the number of samples, research methods, software and other important items in this field.
Introduction:
The term social capital is not well explained in the introduction. Explain it well first, then deal with its divisions.
In my opinion, just stating that there is no longitudinal research in this field is not enough to conduct such research. At the end of the introduction should be a conclusion about what we know, what we do not know. Information gaps in this area should be identified, and the importance of conducting a longitudinal study in this area should be explained to determine the need for research

Experimental design

Method:
Type of study, number of people studied, ethical issues or licenses obtained should be mentioned.
Why are these three scales used? Why are not other scales in this field such as WBA used?
Specify the validity, reliability, and scoring method of each.
On what basis were the 6 questions selected? Have experts in this field selected these items based on their degree of scientific adequacy? Does this scale created from 6 items have validity and reliability for measuring the desired variable?

Validity of the findings

Discussion:
In the discussion section, you can refer to the culture in your community to justify the difference between the results of your research and previous research.
Conclusion:
In the conclusion section, also deal with the cognitive social capital variable

Additional comments

no comment!

Reviewer 2 ·

Basic reporting

The background of the research has not been clearly described. We recommend that researchers add emphasis from the data presented by other researchers regarding the thematic research taken. Research will become sharper and more targeted to recommendations for policymakers and further research development. Researchers should show something unique and different from the study compared to previous studies.

Experimental design

Information about the categories of each variable being studied does not need to be conveyed in the journal narrative. Explanations are simply displayed in tabular form from the research results only. Information about “ethical review” does not need to be displayed in the narrative.

Validity of the findings

In the paper, the author has not written about the weaknesses and limitations found from the research they did.

Additional comments

The writing of bibliographic sources can be corrected again in accordance with the method specified in the publication requirements of scientific journals from PeerJ.

---

## Round 0.2 · accepted · Accept

Many thanks for addressing all the issues.